# Lipid Composition Analysis of Cricket Oil from Crickets Fed with Broken Rice-Derived Bran

**DOI:** 10.3390/insects16090951

**Published:** 2025-09-11

**Authors:** Ryosuke Sogame, Taiki Miyazawa, Masako Toda, Akihiro Iijima, Maharshi Bhaswant, Teruo Miyazawa

**Affiliations:** 1Graduate School of Agricultural Science, Tohoku University, Sendai 980-8572, Miyagi, Japan; taiki.miyazawa.b3@tohoku.ac.jp (T.M.); masako.toda.a7@tohoku.ac.jp (M.T.); 2New Industry Creation Hatchery Center (NICHe), Tohoku University, Sendai 980-8579, Miyagi, Japan; cmaharshi@gmail.com; 3Faculty of Regional Policy, Takasaki City University of Economics, Takasaki 370-0801, Gunma, Japan; a-iijima@tcue.ac.jp; 4FUTURENAUT Inc., Takasaki 370-0801, Gunma, Japan; 5Department of Biotechnology, SRM—Institute of Science and Technology, Tiruchirappalli Campus, Tiruchirappalli 621105, Tamil Nadu, India

**Keywords:** acyl steryl glycoside, broken rice, cricket oil, edible insects, essential fatty acids, fatty acid composition, glycolipid, lipid classes, rice bran, sustainable food

## Abstract

Edible insects have garnered increasing attention due to their high nutritional value, particularly their lipids, which contain a high proportion of ω-6 and ω-3 fatty acids. Among edible insects, the fatty acid composition of cricket oil has been reported to vary depending on the type of feed provided. This study aimed to investigate the effects of rice bran derived from broken rice, a by-product of the rice polishing process, on the lipid composition of cricket oil. Crickets were fed rice bran for seven days, and the resulting changes in the lipid class composition and fatty acid profiles of the extracted oil were analyzed. The results revealed that rice bran supplementation increased the proportion of phospholipids and glycolipids within the total lipid content while exerting a modest effect on the fatty acid composition. These findings suggest that the composition of cricket feed significantly influences the lipid class distribution and fatty acid profiles of cricket oil. This study highlights the potential of crickets as a functional food source, capable of accumulating lipophilic bioactive compounds beneficial to human health. Furthermore, the utilization of agricultural by-products, such as rice bran derived from broken rice, offers a sustainable approach to insect farming while enhancing the nutritional quality of edible insects.

## 1. Introduction

The global population is projected to reach 9.7 billion by 2050 [1], placing increasing pressure on global food systems that are already challenged by climate change. This population growth, coupled with environmental disruptions, may significantly affect food availability and security. Current primary sources of macronutrients, such as fat and protein, are predominantly derived from livestock and aquaculture. However, these industries are associated with substantial environmental impacts, including high water usage, greenhouse gas emissions, and land degradation [2]. In contrast, edible insects have traditionally been consumed in various regions, including Australia, Southeast Asia, and parts of Africa, and are gaining attention as a sustainable and nutritious food source. It is estimated that approximately 2000 species of edible insects are consumed worldwide [3]. Insects are considered environmentally friendly due to their efficient feed conversion ratios, low water and land requirements, and minimal greenhouse gas emission [3]. Nutritionally, insects are rich in proteins, lipids, vitamins, and minerals, including essential amino acids and essential fatty acids such as ω-6 (e.g., linoleic acid) and ω-3 (e.g., α-linolenic acid) [4,5,6,7,8]. The protein content of edible insects varies by species and developmental stages, typically ranging from about 10% to 80% on a dry matter basis [5,9,10,11,12]. Similarly, the lipid content also varies widely, from 10% to 60% of dry matter [5,8,10,12,13]. Due to these nutritional and environmental benefits, edible insects are increasingly being explored as alternative food and feed sources. Furthermore, edible insects are expected to prevent lifestyle-related diseases, such as obesity, which are becoming a global issue, by balancing daily nutrition due to their high nutritional value [12,13].

The global market for edible insects has expanded in recent years. According to the International Platform of Insects for Food and Feed (IPIFF), the production of insect-based food products, such as snacks and nutritional supplement powders, is expected to reach approximately 260,000 tons by 2030 [12,14]. Among edible insects, adult crickets are of particular interest, with lipid contents reported between 10% and 25% on a dry matter basis [6,15,16]. In terms of fatty acid profiles, linoleic acids (a key ω-6 fatty acid) are typically abundant, while oleic acids predominantly represent monounsaturated fatty acids (MUFAs) in crickets [17]. Recent studies have increasingly focused on the impact of feed composition on the nutritional quality of crickets [18,19,20]. For instance, one study reported that the lipid content and composition of Jamaican field crickets (*Gryllus assimilis*) were significantly influenced by dietary lipid sources, which also affected the survival rates and growth performance [21]. Another investigation demonstrated that supplementing house cricket diets with linseed oil, rich in α-linolenic acids, significantly increased the corresponding ω-3 fatty acid levels in the insects [22]. Furthermore, research has shown that feeding crickets with apple cores alters the volatile compound profile of the cricket powder, suggesting that feed has implications for flavor as well as nutrition [23]. Collectively, these findings underscore the importance of feed formulation in modulating the lipid profiles and overall nutritional value of edible insects.

Broken rice, a by-product of rice milling, is often undervalued despite its nutritional equivalence to brown rice. According to the Ministry of Agriculture, Forestry, and Fisheries of Japan, approximately 20 tons of broken rice are generated annually in Japan [24]. Although currently utilized mainly as livestock or pet feed, its high nutritional content suggests untapped potential. Similar trends are observed in other rice-producing regions, where large volumes of broken rice are produced but only minimally consumed [25,26]. In the countries where rice is the staple food, broken rice has an advantage over other agricultural by-products in terms of supply stability. Therefore, exploring the application of broken rice as a feed ingredient for edible insects presents an opportunity for feed cost reduction as well as both waste valorization and sustainable food production.

This study investigates the effects of feeding house crickets with rice bran derived from broken rice on their lipid composition. Specifically, the study focuses on the analysis of lipid classes, including neutral lipids, phospholipids, and glycolipids, as well as the fatty acid composition of cricket-derived oil (Figure 1). By elucidating the relationship between feed composition and the resulting lipid profiles in crickets, this research contributes to the development of nutritionally enhanced and sustainable insect-based food products.

## 2. Materials and Methods

### 2.1. Materials

In this study, oils extracted from adult house crickets (*Acheta domesticus*) by pressing were kindly provided by FUTURENAUT Inc. (Takasaki, Gunma, Japan), a commercial grower. House crickets were reared without separating males and females under two distinct feeding regimes: control feed and feed with rice bran derived from broken rice. The control feed, commonly used for rearing house crickets, primarily consisted of poultry feed, fish meal powder, skim milk, etc. (Figure 2A) [18,27]. Rice bran, derived from broken rice, was kindly provided by NAKARI Corp. (Kami-gun, Miyagi, Japan) and used as the experimental feed (Figure 2B). Crickets in both groups were maintained on the control diet until seven days prior to oil harvest. Thereafter, the control group continued on the same feed, while the experimental group was switched to a rice bran-based diet. On the seventh day, crickets from both groups were harvested, and oils were extracted via mechanical pressing. Prior to the analytical procedures, the crude oils were centrifuged at 1460× *g* to eliminate insoluble particle matter (Figure 2C,D). Because potential effects to prevent growth and to decrease survival ratio was observed with long-term feeding (more than one month of feeding of rice bran) [18], we consequently fed crickets with rice bran for only seven days.

Unless specified otherwise, all other reagents used throughout the study were of special grade or HPLC grade, obtained from FUJIFILM Wako Pure Chemical Corp. (Osaka, Japan).

### 2.2. Extraction of Total Lipids from Cricket Feed

Total lipids from both the control feed and rice bran were extracted using the Folch method [28]. Briefly, 1 g of each feed sample was placed into a 50 mL glass tube and mixed with 16 mL of a chloroform:methanol mixture (2:1, *v*/*v*) containing 0.05% butyl hydroxyl toluene (BHT) as an antioxidant. The samples were incubated overnight at room temperature in the dark to prevent oxidation. Following incubation, 4 mL of an aqueous solution containing 1 mM EDTA and 0.9% NaCl was added, and the mixture was vortexed and centrifuged at 1640× *g* for 10 min. The resulting biphasic solutions were separated, and the lower organic phase containing the lipophilic components was collected, evaporated to dryness, and weighed gravimetrically using an analytical balance (balance XRP205/A, Mettler-Toledo GmbH, Greifensee, Switzerland). The dried lipid extract was redissolved in chloroform:methanol (2:1, *v*/*v*) containing 0.05% BHT to a final concentration of 5 mg/mL and stored at −80 °C until further analysis by one-dimensional thin-layer chromatography (TLC) and gas chromatography-flame ionization detection (GC-FID).

### 2.3. Identification of Lipid Classes by Thin-Layer Chromatography (TLC)

TLC was used to separate and identify the major lipid classes in the cricket oil samples and feed-derived total lipid extracts. TLC silica gel 60 (Merck Millipore, Boston, MA, USA) was used as the stationary phase. For polar lipid separation, the samples were developed using a solvent system of chloroform:methanol:water (75: 25: 4 *v*/*v*/*v*) [29,30]. For non-polar lipid classes, a solvent system of hexane:dimethyl ether: acetic acid (80: 30: 1 *v*/*v*/*v*) was used [30]. The lipid classes were visualized by charring with 40% H_2_SO_4_ followed by heating, enabling spot detection.

### 2.4. Fractionation of Lipid Classes Using Solid-Phase Extraction (SPE)

Lipid fractionation into neutral lipids, glycolipids, and phospholipids was carried out using silica-based solid-phase extraction (SPE), following established protocols [31,32]. SPE cartridge (Sep-Pak Plus Long, pore size of 125 Å, particle size of 55–105 μm, silica weight of 690 mg, Waters Corp., Milford, MA, USA) were preconditioned with chloroform. A 200 μL aliquot of cricket oil (100 mg/mL in chloroform) was loaded onto the column. Elution was performed sequentially with 15 mL of chloroform (neutral lipids), 15 mL of acetone (glycolipids), and 15 mL of methanol (phospholipids). Each fraction was evaporated to dryness and weighed gravimetrically using an electronic balance. The relative proportion of each lipid class was calculated as a percentage of the total lipid mass. Each fraction was subsequently dissolved in chloroform:methanol (2:1) to a final concentration of 5 mg/mL and stored at −80 °C for further analysis using GC-FID.

### 2.5. Analysis of Fatty Acid Composition by GC-FID

The fatty acid composition of total lipids from control feed and rice bran (from Section 2.2), as well as of whole cricket oil and its SPE-derived lipid fractions (from Section 2.4), was determined by GC-FID using a SHIMADZU GC-2010 system (Shimadzu Corp., Kyoto, Japan). The samples were analyzed on a ZB-FAME capillary column (I.D. 0.25 mm, 60.0 m, 0.20 um; Phenomenex Inc., Torrance, CA, USA), following fatty acid methyl ester (FAME) derivatization to previously reported methods [33,34], and heptadecanoic acid was used as an internal standard. Each fatty acid was identified, and their ratio was calculated using FAME-Mix (Sigma Aldrich, St. Louis, MO, USA).

### 2.6. Statistical Analysis

All experimental procedures were performed in independent biological triplicates. Data are presented as a mean ± SEM. The statistical significance was assessed using Welch’s *t*-test in Microsoft Excel (Microsoft Corp., Washington, WA, USA). A *p*-value of <0.05 was considered statistically significant, and a high significance was denoted at *p* < 0.01.

## 3. Results

### 3.1. Identification of Lipid Classes in Cricket Oil and Their Feed

Lipid class profiling of the cricket oil and feed samples was performed using TLC. Consistent with previous reports [12,35], prominent spots corresponding to triacylglycerols (TGAs), phosphatidylethanolamine (PE), and phosphatidylcholine (PC) were detected in cricket oil (Figure 3A). In the low-polarity lipid development system, a spot with an Rf value of 0.38, corresponding to free fatty acids and acyl steryl glycosides, was observed (Figure 3B). This spot was notably more intense in cricket oil from the rice bran-fed group than in the control group, suggesting a higher abundance of these compounds. A similar trend was observed in the corresponding feed samples: rice bran exhibited a more prominent spot than the control feed (Figure 3C), including its richer lipid diversity.

### 3.2. Lipid Class Composition of Cricket Oil

To assess the impact of dietary modifications on lipid class distribution, cricket oils were fractioned by SPE into neutral lipids, glycolipids, and phospholipids and quantified gravimetrically. As shown in Figure 4, the control group oil comprised 87.2% neutral lipids, 11.1% phospholipids, and 1.7% glycolipids. In contrast, the rice bran-fed group exhibited 84.2% neutral lipids, 12.6% phospholipids, and 3.2% glycolipids. The glycolipid fraction was significantly higher in the rice bran-fed group compared to the control (*p* < 0.05), aligning with the TLC results (Figure 3B). A similar increasing trend was observed for phospholipids (*p* = 0.053), indicating a potential influence of rice bran feeding on the polar lipid metabolism in crickets.

### 3.3. Fatty Acid Composition of Control Feed and Rice Bran

Fatty acid profiles of the control feed and the rice bran were analyzed by GC-FID. The control feed predominantly contained palmitic (20.39%), stearic (6.62%), oleic (31.63%), linoleic (22.87%), eicosapentaenoic (EPA) (3.25%), and docosahexaenoic acids (DHAs) (4.34%). On the other hand, rice bran was enriched with palmitic (16.88%), oleic (39.41%), and linoleic acids (37.42%). Notably, rice bran exhibited a markedly higher ω-6/ω-3 fatty acid ratio, approximately 10-fold that of the control feed, due to its high linoleic acid and low ω-3 polyunsaturated fatty acids (PUFAs). The saturated fatty acid (SFA)/unsaturated fatty acid (UFA) ratio was lower in rice bran (0.27) than in the control feed (0.43) (Table 1), aligning with previous characterizations of rice bran and brown rice [36,37,38].

### 3.4. Fatty Acid Composition of Cricket Oil

To determine whether dietary fatty acid profiles influence cricket oil composition, a GC-FID analysis was performed on the whole oil samples. In the control group, the main fatty acids included palmitic (27.70%), stearic (7.71%), oleic (27.77%), and linoleic acids (32.25%), with palmitoleic acid detected at 1.12% (Table 2). The rice bran-fed group showed a similar composition of palmitic (27.04%), oleic (30.00%), and linoleic acids (31.86%) but with a lower palmitoleic acid content (0.85%) (Table 2). The ω-6/ω-3 ratio was significantly higher in the rice bran-fed group than in the control group. Meanwhile, the SFA/UFA ratio was lower in the rice bran-fed group, likely due to its elevated oleic acid content, presumably derived from rice bran (Table 3). Although both groups showed high linoleic acid and palmitic acid contents, the PUFA/MUFA ratio, an important parameter linked to insect development and growth [21], was calculated as 1.15 for the control group and 1.05 for the rice bran-fed group, indicating a subtle dietary impact (Table 3).

### 3.5. Fatty Acid Composition of Neutral Lipid Fractions Extracted from Cricket Oils

A GC-FID analysis was further conducted on neutral lipid fractions. The overall fatty acid composition of the neutral lipid fractions extracted from cricket oils of the control group and rice bran-fed group closely mirrored that of the whole cricket oils (Table 2 and Table 3). Notably, the oleic acid content was slightly elevated, resulting in a lower PUFA/MUFA ratio in both groups (Table 3). Additionally, the ω-6/ω-3 ratios remained consistent with those in their respective whole oils, suggesting that dietary influence on this parameter is maintained in the neutral lipid pool (Table 2 and Table 3).

### 3.6. Fatty Acid Composition of Phospholipid Fraction Extracted from Cricket Oils

The fatty acid profiles of phospholipid fractions revealed a distinct composition compared to other lipid classes (Table 2 and Table 3). Palmitic (control group: 12.25%, rice bran group: 12.60%), stearic (control group: 13.79%, rice bran group: 14.69%), oleic (control group: 15.58%, rice bran group: 15.13%), and linoleic acids (control group: 53.68%, rice bran group: 53.67%) were the major fatty acids detected (Table 2). EPA was present at >1% in both groups (control group: 1.31%, rice bran group: 1.00%) (Table 2). Phospholipids are known to preferentially bind unsaturated fatty acids, particularly at the *sn*-2 position, a characteristic that has also been reported in the internal environment of crickets [39,40]. As reported in a previous study, the SFA/UFA ratio in the phospholipid fraction in cricket oil was lower than that of other lipid fractions [12]. Interestingly, despite the high oleic acid content in the feed, MUFA levels in phospholipids were relatively low. Furthermore, no significant difference was observed in the PUFA/MUFA ratio between both groups, a trend that was not observed in other fractions as well as total lipid (Table 3). Additionally, the ω-6/ω-3 ratio in both groups was similar to that of their respective whole cricket oil (Table 3).

### 3.7. Fatty Acid Composition of Glycolipid Fraction Extracted from Cricket Oils

Glycolipid fractions also reflected notable dietary influences (Table 2 and Table 3). Palmitic (control group: 23.31%, rice bran-fed group: 25.83%), stearic (control group: 9.55%, rice bran-fed group: 7.92%), oleic (control group: 26.80%, rice bran-fed group: 26.67%), and linoleic acids (control group: 34.56%, rice bran-fed group: 35.98%) were the predominant fatty acids in the glycolipid fraction (Table 2). The ω-6/ω-3 ratio in the rice bran-fed group was consistent with that of its respective whole cricket oil (Table 3). However, the control feed-fed group showed a significantly lower ω-6/ω-3 ratio. The MUFA/PUFA ratio was higher in the glycolipids than in the whole oil, particularly in the control group (Table 2). When comparing the major fatty acids between the control and rice bran groups, significant differences were observed for all fatty acids except oleic acid. Furthermore, palmitic acid was more abundant than stearic acid in glycolipids, whereas both were nearly equal in phospholipids (Table 2 and Table 3), suggesting differential partitioning of SFAs among lipid classes.

## 4. Discussion

Among the wide array of edible insects, house crickets (*Acheta domesticus*) have emerged as one of the most prominent species in the global edible insect market. Their popularity stems not only from their relatively high reproductive efficiency and ease of farming but also from their widespread acceptance and consumption across multiple cultures and regions [41]. As the edible insect industry expands, understanding how dietary factors influence the nutritional quality of insects is becoming increasingly important. Previous research has demonstrated that the composition of insect feed is a critical determinant of the nutritional profile of the insects themselves, particularly in terms of lipid content and fatty acid composition. Insects are known to incorporate dietary lipids into their tissues, and such incorporation can affect physiological parameters, including survival rates, body mass, and adult size metrics, such as body length [19,20,21,27].

In this context the present study represents a novel investigation into the effect of incorporating broken rice-derived bran—an abundant agricultural by-product—into the diets of house crickets. This approach aligns with ongoing global efforts to improve food sustainability by repurposing waste materials and underutilized resources in agriculture. Our findings provide new insights into how such feed alterations may influence the lipid class distribution and fatty acid profile of cricket oil. In this study, we conducted an analysis of lipid classes and fatty acid composition as a first step toward understanding the effects of rice bran feed on the nutritional components of crickets, because these analyses had been performed in several previous studies focused on the effects of cricket feed [19,20,21,22].

The results demonstrated that, while the overall major lipid classes were similar between the control group and the rice bran-fed group, there was a notable increase in the glycolipid fraction in the rice bran-fed group, as identified through both thin-layer chromatography (TLC) and solid-phase extraction (SPE). This increase is likely attributable to the glycolipid-rich nature of rice bran [36,38,39], which is known to contain a diverse array of glycolipids, including diglycosyl diglycerides, monoglycosyl monoglycerides, glucosyl ceramides, steryl glycosides, and acyl steryl glycosides—of which the latter is particularly abundant [36,38,39]. Acyl steryl glycosides are characterized by their low polarity and exhibit chromatographic behavior similar to that of free fatty acids. However, we could not fully identify the increase in glycolipids in this study. Therefore, we believe that a more detailed analysis of glycolipids is desirable in the future.

While some previous studies have reported substantial changes in the fatty acid profiles of crickets based on dietary variations [21,22], the present study found relatively modest alterations. This discrepancy may be due to the similarity of fatty acid composition between rice bran and crickets rather than the shorter feeding period employed in our experimental design, which lasted approximately seven days. As in previous studies, oleic acid, followed by linoleic acid, was most abundant in this study, and these two fatty acids are also most abundant in the control feed. These similarities shown in fatty acid composition between the two different types of feed may have contributed to the small change in fatty acid composition in the crickets. Supporting this, the fatty acids such as eicosapentaenoic acid and docosahexaenoic acid, which are not present in rice bran, are decreased in the cricket oil of the rice bran group. These invariances were shown in a previous study in which crickets were fed with apple cores containing oleic acid and linoleic acid as predominant fatty acids for seven days [23,40], although other fatty acids that are not present in apple cores, volatile compounds, and sugars were significantly changed [23]. However, longer exposure to modified diets may elicit more pronounced shifts in fatty acid composition [19,20,21]. Nonetheless, a statistically significant increase in glycolipids and slight differences in phospholipid content between the groups were observed, suggesting some degree of lipid remodeling in response to the rice bran diet. With respect to fatty acid composition, feeding crickets with rice bran resulted in increased ω-6/ω-3 polyunsaturated fatty acid (PUFA) ratios across all lipid fractions. The oleic acid in the total lipids of the rice bran feed group showed the greatest increase compared to the control group, but the increase was less than 2%. However, this trend was observed only in the neutral lipid fraction and not in the phospholipid and glycolipid fraction. Additionally, distinctive changes of fatty acid profiles within the two groups could not be observed in the phospholipid fractions and glycolipid fractions, although the proportions of both fractions within the total lipids were different in the two groups. A previous study reported that house crickets can produce linoleic acid from oleic acid by Δ12-desaturase and elongate linoleic acid to arachidonic acid [41]. This may indicate that house crickets regulate the fatty acid content in phospholipids and glycolipids. An elevated ω-6/ω-3 ratio, while common in many Western diets, is generally considered less desirable from a human health perspective. However, the ability to manipulate such ratios through controlled feeding could allow for the targeted modulation of lipid profiles in edible insects, depending on the intended nutritional application [42,43,44].

Although this study was limited to a relatively short feeding duration, it suggests the potential of rice bran as a functional feed ingredient for modifying lipid composition in crickets. Future investigations should explore longer-term feeding strategies to assess whether the observed changes are sustained or further amplified over time. Moreover, a comprehensive understanding of the bioavailability, metabolic fate, and physiological roles of individual lipid species within the insect body is essential. This includes determining how these lipids behave upon human consumption, whether in the form of extracted cricket oil or whole-powder cricket-based products. For example, stearic acid, as one of the saturated fatty acids, has been shown to have a positive effect on human health [45]. Such research could inform the development of tailored insect-derived foods with specific health benefits. Furthermore, research focusing on such feed will require consideration of costs and yields in order to achieve sustainability.

## 5. Conclusions

In response to the growing demand for sustainable and nutritionally functional food sources, this study demonstrated that the inclusion of rice bran—derived from the agricultural by-product broken rice—into cricket feed can lead to several notable changes in the lipid composition of house crickets. In this study, we could not find obvious differences in the fatty acid composition between both cricket oils. However, the glycolipid content of cricket oil was significantly increased following rice bran supplementation, without great changes in fatty acid composition.

These findings underscore the feasibility of using agricultural by-products to enhance the functional lipid profile of edible insects. As cricket oil may serve as a novel and sustainable source of dietary lipids for human consumption, understanding how diet influences its composition is crucial for optimizing its nutritional quality. Nevertheless, further studies are necessary to validate the long-term effects of such dietary interventions and to investigate the health implications and bioactive properties of the modified lipid constituents. Such research will be instrumental in establishing cricket oil as a viable functional food ingredient within the broader landscape of sustainable nutrition.

## Figures and Tables

**Figure 1 insects-16-00951-f001:**
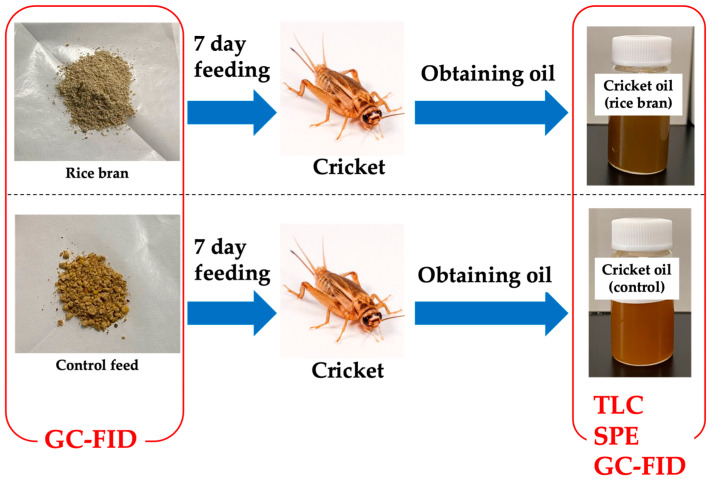
Scheme of the present study. GC-FID, gas chromatography-flame ionization detection; SPE, solid-phase extraction; TLC, thin-layer chromatography.

**Figure 2 insects-16-00951-f002:**
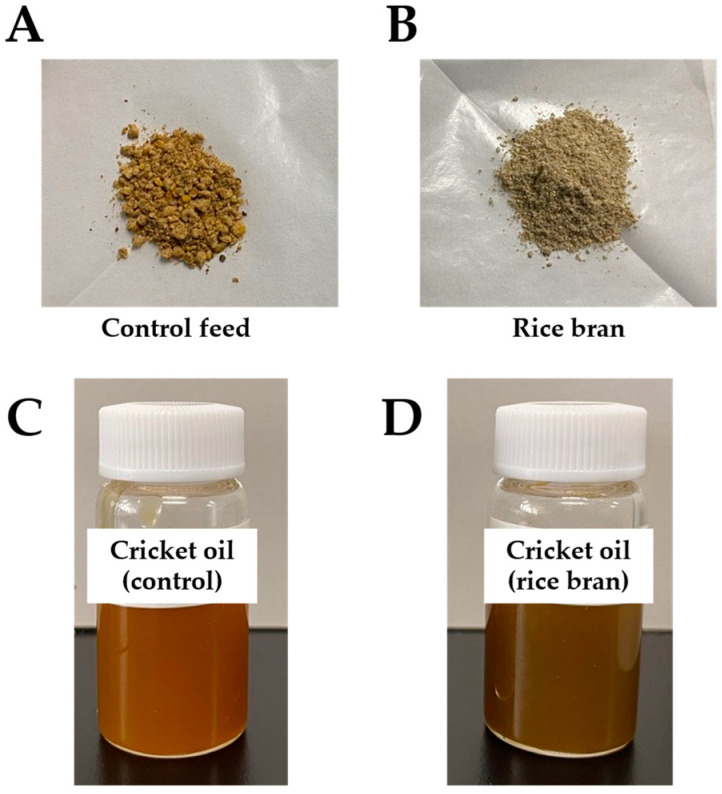
External appearance of the samples used in this study. Photographs of the cricket feed: (**A**) control feed and (**B**) rice bran. Photographs of cricket oil (*Acheta domesticus*) after centrifugation (1460 *g*) obtained from two different feeding methods: (**C**) control feed-fed group and (**D**) rice bran-fed group.

**Figure 3 insects-16-00951-f003:**
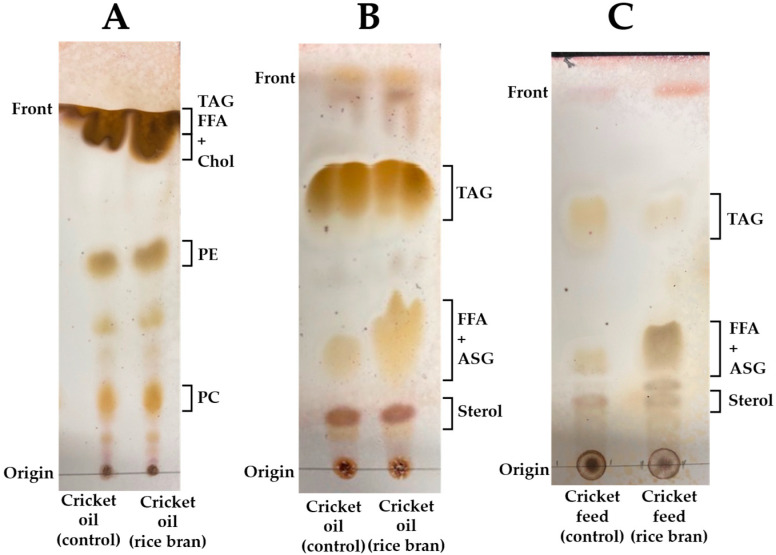
Photograph of the development of lipids in each group by one-dimensional thin-layer chromatography (TLC). (**A**) Cricket oil developed with chloroform: methanol: water (75:25:4 *v*/*v*/*v*). (**B**) Cricket oil developed with hexane: dimethyl ether: acetic acid (80:30:1 *v*/*v*/*v*). (**C**) Cricket feed developed with hexane: dimethyl ether: acetic acid (80:30:1 *v*/*v*/*v*). ASG, acyl steryl glycoside; Chol, cholesterol; FFA, free fatty acid; PE, phosphatidylethanolamine; PC, phosphatidylcholine; Sterol, sterol lipids; TAG, triacylglycerol.

**Figure 4 insects-16-00951-f004:**
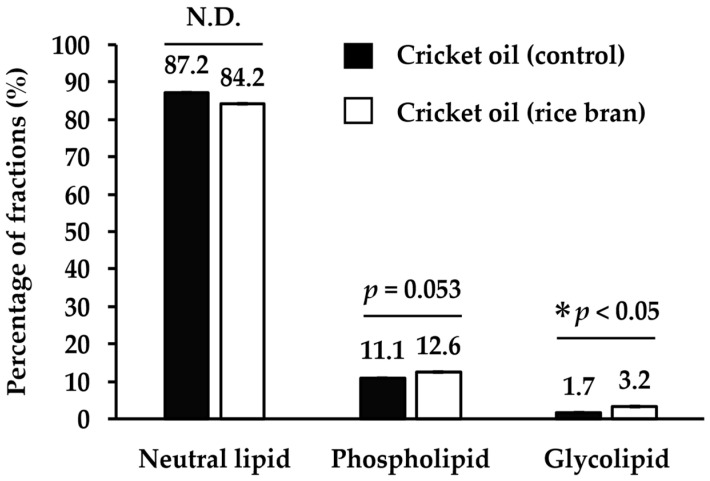
Percentages of neutral lipid, phospholipid, and glycolipid fractions in cricket oil. Values are expressed as means ± SEM of triplicate analyses. The percentage of each lipid fraction was calculated using the sum of all lipid fraction weights as 100%. An asterisk (*) indicates statistically significant differences (Student’s *t*-test, *p* < 0.05). N.D., not detected.

**Table 1 insects-16-00951-t001:** Fatty acid composition of total lipids extracted from cricket feeds.

	Cricket Feed (Control)	Cricket Feed (Rice Bran)
C12:0 (Lauric acid)	0.04 ± 0.01	N.D.
C14:0 (Myristic acid)	1.69 ± 0.12	0.35 ± 0.01
C15:0 (Pentadecanoic acid)	0.22 ± 0.01	0.07 ± 0.00
C16:0 (Palmitic acid)	20.39 ± 0.14	16.88 ± 0.08
C16:1 (cis-9 Palmitoleic acid	2.67 ± 0.09	0.18 ± 0.01
C18:0 (Stearic acid)	6.62 ± 0.05	1.96 ± 0.00
C18:1 (trans-9 Elaidic acid)	0.63 ± 0.02	N.D.
C18:1 (cis-9 Oleic acid)	31.63 ± 0.32	39.41 ± 0.05
C18:2 (cis-9,12 Linoleic acid)	22.87 ± 0.52	37.42 ± 0.05
C18:3 (cis-9,12,15 α-Linolenic acid)	1.36 ± 0.02	1.40 ± 0.00
C20:0 (Arachidic acid)	0.35 ± 0.00	0.58 ± 0.00
C20:1 (cis-11 Eicosenoic acid)	1.90 ± 0.24	0.47 ± 0.01
C20:2 (cis-11,14 Eicosadienoic acid)	0.15 ± 0.01	0.02 ± 0.00
C20:4 (cis-5,8,11,14 Arachidonic acid)	0.46 ± 0.01	N.D.
C20:5 (cis-5,8,11,14,17 Eicosapentaenoic acid)	3.25 ± 0.11	N.D.
C22:0 (Behenic acid)	0.30 ± 0.15	0.39 ± 0.00
C22:1 (cis-13 Erucic acid)	0.31 ± 0.02	0.02 ± 0.01
C22:6 (cis-4,7,10,13,16,19 Docosahexaenoic acid)	4.34 ± 0.16	N.D.
C23:0 (Tricosanoic acid)	0.03 ± 0.00	0.06 ± 0.01
C24:0 (Lignoceric acid)	0.28 ± 0.00	0.76 ± 0.00
C24:1 (Nervonic acid)	0.29 ± 0.05	0.01 ± 0.00

SFA ratio	29.93 ± 0.33	21.05 ± 0.08
MUFA ratio	37.49 ± 0.09	40.10 ± 0.04
PUFA ratio	32.58 ± 0.29	38.85 ± 0.05
ω-6 ratio	23.58 ± 0.51	37.45 ± 0.05
ω-3 ratio	9.00 ± 0.25	1.40 ± 0.00
ω-6/ω-3 ratio	2.63 ± 0.13	26.69 ± 0.04
SFA/UFA ratio	0.43 ± 0.01	0.27 ± 0.00
PUFA/MUFA ratio	0.87 ± 0.01	0.97 ± 0.00

Data are expressed as % of total lipid and as means ± SEM of triplicate analyses. N.D., not detected.

**Table 2 insects-16-00951-t002:** Fatty acid composition of cricket oils and their lipid fractions.

	Total Lipid		Neutral Lipid Fraction		Phospholipid Fraction		Glycolipid Fraction	
	Cricket Oil (Control Feed)	Cricket Oil (Rice Bran)	Cricket oil (Control Feed)	Cricket oil (Rice Bran)	Cricket Oil (Control Feed)	Cricket Oil (Rice Bran)	Cricket Oil (Control Feed)	Cricket Oil (Rice Bran)
C12:0 (Lauric acid)	0.1 ± 0	0.1 ± 0 **	0 ± 0	0 ± 0	ND	ND	0 ± 0	0 ± 0
C14:0 (Myristic acid)	0.9 ± 0	0.8 ± 0 **	0.6 ± 0.1	0.6 ± 0.1	0.1 ± 0	0.1 ± 0	0.6 ± 0	0.5 ± 0 *
C14:1 (cis-9 Myristoleic acid)	0 ± 0	0 ± 0	0 ± 0	0 ± 0	N.D.	N.D.	N.D.	N.D.
C15:0 (Pentadecanoic acid)	0.1 ± 0	0.1 ± 0	0.1 ± 0	0.1 ± 0	0.1 ± 0	0.1 ± 0	0.2 ± 0	0.1 ± 0 *
C16:0 (Palmitic acid)	28 ± 0.1	27 ± 0 **	28 ± 0.2	27 ± 0.2	12 ± 0.1	13 ± 0.2	23 ± 0.2	26 ± 0.2 **
C16:1 (cis-9 Palmitoleic acid)	1.1 ± 0	0.9 ± 0 **	1.1 ± 0	0.8 ± 0 **	0.5 ± 0	0.4 ± 0	1.1 ± 0	0.8 ± 0 **
C18:0 (Stearic acid)	7.7 ± 0	7.4 ± 0 **	7.3 ± 0	6.9 ± 0 **	14 ± 0.1	15 ± 0.1 **	9.6 ± 0.1	7.9 ± 0 **
C18:1 (trans-9 Elaidic acid)	0.3 ± 0	0.3 ± 0	0.5 ± 0	0.3 ± 0 *	0.6 ± 0	0.6 ± 0	0.5 ± 0	0.3 ± 0 *
C18:1 (cis-9 Oleic acid)	28 ± 0	30 ± 0 **	29 ± 0.1	32 ± 0.1 **	16 ± 0.1	15 ± 0	27 ± 0.2	27 ± 0.1
C18:2 (trans-9,12 Linolelaidic acid)	N.D.	N.D.	N.D.	N.D.	0 ± 0	0 ± 0	0.1 ± 0	0 ± 0
C18:2 (cis-9,12 Linoleic acid)	32 ± 0	32 ± 0 **	31 ± 0.2	30 ± 0.1 *	54 ± 0.2	54 ± 0.1	35 ± 0.3	36 ± 0.2 *
C18:3 (cis-9,12,15 Linolenic acid)	0.7 ± 0	0.5 ± 0 **	0.7 ± 0	0.5 ± 0 **	0.4 ± 0	0.3 ± 0 **	0.7 ± 0	0.4 ± 0 **
C20:0 (Arachidic acid)	0.3 ± 0	0.3 ± 0	0.3 ± 0	0.3 ± 0	0.3 ± 0	0.4 ± 0 *	0.5 ± 0	0.4 ± 0
C20:1 (cis-11 Eicosenoic acid)	0.1 ± 0	0.1 ± 0	0.1 ± 0	0.1 ± 0 *	0.1 ± 0	0 ± 0 *	0.3 ± 0	0.1 ± 0 **
C20:2 (cis-11,14 Eicosadienoic acid)	0 ± 0	0 ± 0 *	0 ± 0	0 ± 0	0.1 ± 0	0.1 ± 0 *	0.1 ± 0	0 ± 0 **
C20:3 (cis-8,11,14 Eicosatrienoic acid)	N.D.	N.D.	N.D.	N.D.	0 ± 0	0 ± 0	N.D.	N.D.
C20:4 (cis-5,8,11,14 Arachidonic acid)	0.2 ± 0	0.1 ± 0 **	0.1 ± 0	0.1 ± 0 *	1 ± 0	0.7 ± 0 **	0.3 ± 0	0.2 ± 0 **
C20:5 (cis-5,8,11,14,17 Eicosapentaenoic acid)	0.4 ± 0	0.3 ± 0 **	0.3 ± 0	0.2 ± 0 **	1.3 ± 0.1	1 ± 0.1 *	0.6 ± 0	0.3 ± 0 **
C22:0 (Behenic acid)	0 ± 0	0 ± 0	0 ± 0	0 ± 0	N.D.	N.D.	0.2 ± 0	0.1 ± 0
C22:1 (cis-13 Erucic acid)	0 ± 0	0 ± 0	0 ± 0	0 ± 0	0 ± 0	0 ± 0	0.1 ± 0	0 ± 0 **
C22:6 (cis-4,7,10,13,16,19 Docosahexaenoic acid)	0.1 ± 0	0 ± 0 **	0.1 ± 0	0 ± 0 **	0.1 ± 0	0 ± 0 **	0.2 ± 0	0 ± 0 **
C23:0 (Tricosanoic acid)	0.2 ± 0	0.3 ± 0 **	0.2 ± 0	0.3 ± 0 **	0.1 ± 0	0.1 ± 0	0.1 ± 0	0.1 ± 0.1
C24:0 (Lignoceric acid)	0 ± 0	0 ± 0	0 ± 0	0 ± 0	0 ± 0	0 ± 0	0.3 ± 0	0.1 ± 0 *

Data are expressed as % of total lipid and as means ± SEM of triplicate analyses. An asterisk (*) indicates statistically significant differences (Student’s *t*-test, * *p* < 0.05, ** *p* < 0.01). N.D., not detected.

**Table 3 insects-16-00951-t003:** Fatty acid ratio of cricket oils and their lipid fractions.

	Total Lipid		Neutral Lipid Fraction		Phospholipid Fraction		Glycolipid Fraction	
	Cricket Oil (Control Feed)	Cricket Oil (Rice Bran)	Cricket Oil (Control Feed)	Cricket Oil (Rice Bran)	Cricket Oil (Control Feed)	Cricket Oil (Rice Bran)	Cricket Oil (Control Feed)	Cricket Oil (Rice Bran)
SFA	37 ± 0.1	36 ± 0 **	37 ± 0.3	35 ± 0.2 *	27 ± 0.1	28 ± 0.1 **	35 ± 0.2	35 ± 0.1 *
MUFA	29 ± 0.1	31 ± 0 **	31 ± 0.1	33 ± 0.1 **	17 ± 0.1	16 ± 0 *	29 ± 0.1	28 ± 0.1 *
PUFA	34 ± 0	33 ± 0 **	32 ± 0.2	31 ± 0.1 **	57 ± 0.3	56 ± 0.1	37 ± 0.3	37 ± 0.2
ω-6	33 ± 0	32 ± 0 **	31 ± 0.2	31 ± 0.1 *	55 ± 0.3	55 ± 0.1	35 ± 0.3	36 ± 0.2 *
ω-3	1.1 ± 0	0.8 ± 0 **	1.1 ± 0	0.7 ± 0 **	1.7 ± 0.1	1.3 ± 0.1 *	1.6 ± 0	0.8 ± 0 **
ω-6/ω-3	29 ± 0.1	41 ± 0.1 **	29 ± 0	41 ± 0.6 **	32 ± 1.1	43 ± 2.7 *	22 ± 0.2	46 ± 0.8 **
SFA/UFA	0.6 ± 0	0.6 ± 0 **	0.6 ± 0	0.6 ± 0 *	0.4 ± 0	0.4 ± 0 **	0.5 ± 0	0.5 ± 0
PUFA/MUFA	1.2 ± 0	1.1 ± 0 **	1 ± 0	0.9 ± 0 **	3.4 ± 0	3.5 ± 0	1.3 ± 0	1.3 ± 0 *

Data are expressed as % of total lipid or each fraction and as means ± SEM of triplicate analyses. An asterisk (*) indicates statistically significant differences (Student’s *t*-test, * *p* < 0.05, ** *p* < 0.01).

## Data Availability

The original contributions presented in the study are included in the article, further inquiries can be directed to the corresponding authors.

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
