# Peer review of "Lipid Composition Analysis of Cricket Oil from Crickets Fed with Broken Rice-Derived Bran"

_insects, 2025, doi:10.3390/insects16090951_

Round 1
Reviewer 1 Report
Comments and Suggestions for Authors
It is recommended that the authors analyze the potential effects of the experimental diet on cricket growth performance. Specifically, considering that the feeding period using the formulated diet was only 7 days, it is worth questioning whether this duration is sufficient to induce significant changes in oil quality. Additional references supporting the impact of short-term dietary interventions on lipid composition in insects would strengthen the discussion.
Author Response
Dear Reviewer 1:
Thank you for reviewing our manuscript. the comment from reviewer 1 is very important and essential for this manuscript. Thank you again for spending your valuable time on this manuscript. We made some changes to this manuscript according to your comment and we would you like to confirm the below revised comment and manuscript.
Comment 1: It is recommended that the authors analyze the potential effects of the experimental diet on cricket growth performance. Specifically, considering that the feeding period using the formulated diet was only 7 days, it is worth questioning whether this duration is sufficient to induce significant changes in oil quality. Additional references supporting the impact of short-term dietary interventions on lipid composition in insects would strengthen the discussion.
Response: Thank you for the comment. We fed rice bran crickets seven days in this study, because it was suggested that long-term of rice bran feeding markedly decreased the survival ratio and lengthened the development time in previous study (https://doi.org/10.1371/journal.pone.0313083). Moreover, another previous study (https://doi.org/10.3390/foods13111668) which fed crickets with apple cores, a byproduct of apple, for seven days. In this study, short-term feeding method was sufficient to vary the fatty acid composition in the crickets. Therefore, we decided seven days to feed rice bran to crickets in this study. However, we couldn’t observe any obvious differences between two groups. We thought that it is due to the similarities of fatty acid composition in the control diets and the rice bran, therefore, we are planning to investigate the influence of rice bran feeding on properties of cricket oil more deeply by modifying the feeding method, such as to lengthen period in our future work.
We added the following description of the reason we had chosen 7 days for feeding period in Materials and Methods section 2.1. (P3, L28 to 30, in revised manuscript).
“Because potential effects to prevent growth and to decrease survival ratio was observed on long-term more than one month feeding of rice bran in the previous study [17], we consequently fed cricket with rice bran for only seven days.”
Reviewer 2 Report
Comments and Suggestions for Authors
Title: Composition analysis of cricket oil from crickets with broken rice-derived bran
Authors: Sogame R. et al.
The task that the authors have given themselves is good and the research as well as the English of the ms is generally acceptable. However, there a number of inaccuracies, important key papers missing that would have needed to be referred to, and some not so usual approaches
Let’s start with the still commonly repeated statement that the global population is rapidly growing. That is simply not a fact anymore (maybe it was when Meyer-Rochow V.B. in 1975 was the first to publish a paper titled “Can Insects Help To Ease The Problem Of World Food Shortage?” in Search 6(7), 261-262 and suggested that WHO and FAO support the idea of using insects as good), but nowadays the dramatic population decreases seen in almost all Western countries and East Asian nations are alarming and recently even China reported a decreasing population and India is heading that way, too. I mention that because authors have a duty to see the situation critically and more differentiated, otherwise they are misleading the readership. Those western countries with a small population increase obtain the extra population through immirants!
The problem these days is an “overproduction of food” and the resultant obesity which is a problem so widespread that health officials and population scientists are seriously worried - even in urban India and China. Health deterioration due to obesity (which ‘kills’ far, far more than a lack of food!) has been widely publicized and can no longer be ignored. But what CAN be said is that many people are not consuming nutritionally balanced food but gorge themselves with junk food. WHO figures show: In 2022, 1 in 8 people in the world were living with obesity. Worldwide adult obesity has more than doubled since 1990, and adolescent obesity has quadrupled. In 2022, 2.5 billion adults (18 years and older) were overweight. Of these, 890 million were living with obesity. https://www.who.int/news-room/fact-sheets/detail/obesity-and-overweight#:~:text=In%202022%2C%201%20in%208,million%20were%20living%20with%20obesity.
Another ‘wrong’: It is shocking that this totally insane and inaccurate figure of 2 billion insect-consuming people is still being cited! NEVER cite such inaccurate figures in a scientific paper! Vastly inflated, this grossly erroneous statement is based on, for example, a tiny number of people in Japan from the mountainous provinces of Gifu and Nagano, who very occasionally consume wasps and grasshoppers (perhaps less than 1 million people), which then used to claim that all 120 million Japanese regular eat insects. Pureb nonsense. (I lived in Japan for many years and speak Japanese). Or does it justify to include 53 million Koreans as insect consumers, when perhaps a few mostly elderly citizens very occasionally buy and eat ‘bondaegi’ (silkworms)? I have a Korean wife. Honestly, that figure of 2 billion was ‘invented’ by people with a vested interest to promote insects as food, especially crickets, in Europe and it is simply wrong and misleading to cite it. Delete the citation or refer to “Arnold Van Huis, A. Van Halloran, Joost Van Itterbeeck 2022: “…the much cited 2 billion figure in the FAO/WUR report must be an overestimation.” In: How many people on our planet eat insects: 2 billion? Journal of Insects as Food and Feed 8(1):1-4. Besides, what does it mean anyhow, when you read, so-and-so many people consume such-and-such numbers of insects? How often: per day, month, year, in a lifetime….?
However, the authors are correct with this statement: insect compositions and nutritional value depend on many factors and the paper by Meyer-Rochow et al. 2021 “Chemical Composition, Nutrient Quality and Acceptability of Edible Insects Are Affected by Species, Developmental Stage, Gender, Diet, and Processing Method”. Foods, https://doi.org/10.3390/foods10051036, should have been read and cited in this context! For fatty acid analyses several important original papers by S. Ghosh et al in the Journal of Asia-Pacific Entomology contain detailed analysis and experimental data on insect lipids and fatty acids (not simply copied from other papers). Incidentally, not all saturated fatty acids are bad, since it has been reported in some of the world’s top medical journals that (at least stearic acid) has a positive effect on human health, see Bonanome, A., Grundy, S.M., 1988. Effect of dietary stearic acid on plasma cholesterol and lipoprotein levels. N. Engl. J. Med. 318, 1244–1248.
Why do the authors write “This study aims to…”? Why not write in a more positive way such “This study investigates…” or “This study shows that…”? Are the authors not certain whether their results are correct, so that they can only ‘aim’? In their Methods the information on the material they used is deficient: what were the amounts of cricket used for control and experimental analyses? Were juveniles and adults together analysed? And were males and females separated? This would have been quite important as sexual differences in the nutritional value between male and female house crickets exist, with females containing significantly higher amounts of lipids (but less protein and chitin) than males, has been reported by Kulma et al. (2018): Kulma M, Kourinska L, Plachy V, Bozik M, Adamkova A, Vrabec V (2019): Effect of sex on the nutritional value of house cricket Acheta domestica. Food Chem 272: 267-272.
Finally, I realise that in their lipid analyses, the authors distinguish phosphatidylethanolamine (PE), triacylglycerols (TGA), and phosphatidylcholine (PC) as well as free fatty acids. Why not also SM, PS, and PI, after all Van Meer, G 1989 in “Lipid traffic in animal cells. Ann. Rev. Cell. Biol. 5, 247-255”, already had pointed out (and after him many more investigators) that lipids of all membranes contain phosphatidylcholine PC, phosphatidylethanolamine PE, sphingomyelin SM, phosphatidylserine PS, and phosphatidylinositol PI ?
The authors did see a significant difference with regard to glycolipids, which were significantly increased in the rice bran fed crickets, although with 3.2% of all lipids still trailing far behind the others. What would the differences mean in connection with the health of the consumer (animal or human)?
Although, I’ve listed a number of problems, I would like to see the paper published, provided the authors can accommodate my concerns, cite the missing key papers and once again check their English (already on Line 2 of the Summary it should be “…lipids which contain a…”
Incidentally, it seems a little daft to have a “Simple Summary” and an “Abstract”, which basically contain the same information. Ask the journal’s chief editor what s/he wants to have in the Simple Summary and what in the Abstract. To me it’s not clear.
Comments on the Quality of English Languageyou have the occasional problem with the use of "a / an" and "the" as well as singularb and plural. Please check your ms.
Author Response
The task that the authors have given themselves is good and the research as well as the English of the ms is generally acceptable. However, there a number of inaccuracies, important key papers missing that would have needed to be referred to, and some not so usual approaches
Dear Reviewer 2
Thank you for reviewing our manuscript. All the comments from reviewer 2 are very important and essential for this manuscript. Thank you again for spending your valuable time on this manuscript. We made changes to this manuscript according to suggestions from reviewer 2, and we would you like to confirm the below revised comments and manuscript.
Comment 1: Let’s start with the still commonly repeated statement that the global population is rapidly growing. That is simply not a fact anymore (maybe it was when Meyer-Rochow V.B. in 1975 was the first to publish a paper titled “Can Insects Help To Ease The Problem Of World Food Shortage?” in Search 6(7), 261-262 and suggested that WHO and FAO support the idea of using insects as good), but nowadays the dramatic population decreases seen in almost all Western countries and East Asian nations are alarming and recently even China reported a decreasing population and India is heading that way, too.
Response: Thank you for the comment. We deleted the word “rapid” in the introduction section (P2, L3, in revised manuscript).
Comment 2: The problem these days is an “overproduction of food” and the resultant obesity which is a problem so widespread that health officials and population scientists are seriously worried - even in urban India and China. Health deterioration due to obesity (which ‘kills’ far, far more than a lack of food!) has been widely publicized and can no longer be ignored. But what CAN be said is that many people are not consuming nutritionally balanced food but gorge themselves with junk food. WHO figures show: In 2022, 1 in 8 people in the world were living with obesity. Worldwide adult obesity has more than doubled since 1990, and adolescent obesity has quadrupled. In 2022, 2.5 billion adults (18 years and older) were overweight. Of these, 890 million were living with obesity. https://www.who.int/news-room/fact-sheets/detail/obesity-and-overweight#:~:text=In%202022%2C%201%20in%208,million%20were%20living%20with%20obesity.
Response: Thank you for the comment. we agree with your opinion, so we added the following description regarding lifestyle-related diseases into Introduction section (P2, L22 to 24, in revised manuscript).
“Furthermore, edible insects are expected to prevent lifestyle-related diseases such as obesity, which are becoming a global issue, by balancing daily nutrition due to their high nutritional value[12, 13].”
Comment 3: Another ‘wrong’: It is shocking that this totally insane and inaccurate figure of 2 billion insect-consuming people is still being cited! NEVER cite such inaccurate figures in a scientific paper! Vastly inflated, this grossly erroneous statement is based on, for example, a tiny number of people in Japan from the mountainous provinces of Gifu and Nagano, who very occasionally consume wasps and grasshoppers (perhaps less than 1 million people), which then used to claim that all 120 million Japanese regular eat insects. Pureb nonsense. (I lived in Japan for many years and speak Japanese). Or does it justify to include 53 million Koreans as insect consumers, when perhaps a few mostly elderly citizens very occasionally buy and eat ‘bondaegi’ (silkworms)? I have a Korean wife.
Response: Thank you for the comment. We delete the statement “over two billion people consume.” (P2, L12, in revised manuscript)
Comment 4: However, the authors are correct with this statement: insect compositions and nutritional value depend on many factors and the paper by Meyer-Rochow et al. 2021 “Chemical Composition, Nutrient Quality and Acceptability of Edible Insects Are Affected by Species, Developmental Stage, Gender, Diet, and Processing Method”. Foods, https://doi.org/10.3390/foods10051036, should have been read and cited in this context! For fatty acid analyses several important original papers by S. Ghosh et al in the Journal of Asia-Pacific Entomology contain detailed analysis and experimental data on insect lipids and fatty acids (not simply copied from other papers). Incidentally, not all saturated fatty acids are bad, since it has been reported in some of the world’s top medical journals that (at least stearic acid) has a positive effect on human health, see Bonanome, A., Grundy, S.M., 1988. Effect of dietary stearic acid on plasma cholesterol and lipoprotein levels. N. Engl. J. Med. 318, 1244–1248.
Response: Thank you for the comment. We added the article (https://doi.org/10.3390/foods10051036) in Reference42.
Comment 5: Why do the authors write “This study aims to…”? Why not write in a more positive way such “This study investigates…” or “This study shows that…”? Are the authors not certain whether their results are correct, so that they can only ‘aim’?
Response: Thank you for the comment. We modified the description in the introduction section (P3, L4, in revised manuscript).
Comment 6: In their Methods the information on the material they used is deficient: what were the amounts of cricket used for control and experimental analyses? Were juveniles and adults together analysed? And were males and females separated? This would have been quite important as sexual differences in the nutritional value between male and female house crickets exist, with females containing significantly higher amounts of lipids (but less protein and chitin) than males, has been reported by Kulma et al. (2018): Kulma M, Kourinska L, Plachy V, Bozik M, Adamkova A, Vrabec V (2019): Effect of sex on the nutritional value of house cricket Acheta domestica. Food Chem 272: 267-272.
Response: Thank you for the comment. We added the details of cricket oils into Materials and Methods 2.1 Material (P2, L19 to 20, in revised manuscript).
“In this study, oils extracted from adult house crickets (Acheta domesticus) by pressing were kindly provided by FUTURENAUT Inc. (Gunma, Japan), a commercial grower. House crickets were reared without separating males and females under two distinct feeding regimes: control feeds and rice brans derived from broken rice. The control feed, commonly used for rearing house crickets, primarily consisted of poultry feed, fish meal powder, skim milk, etc. (Fig. 2 A) [17,26].”
Comment 7: Finally, I realise that in their lipid analyses, the authors distinguish phosphatidylethanolamine (PE), triacylglycerols (TGA), and phosphatidylcholine (PC) as well as free fatty acids. Why not also SM, PS, and PI, after all Van Meer, G 1989 in “Lipid traffic in animal cells. Ann. Rev. Cell. Biol. 5, 247-255”, already had pointed out (and after him many more investigators) that lipids of all membranes contain phosphatidylcholine PC, phosphatidylethanolamine PE, sphingomyelin SM, phosphatidylserine PS, and phosphatidylinositol PI ?
Response: Thank you for the comment. In this study, we identified only PC and PE as phospholipid because other standards such you mentioned were high cost. However, we think focusing on other phospholipids as well as glycolipid is very important, thus we intend to conduct research more deeply in the future.
Comment 8: The authors did see a significant difference with regard to glycolipids, which were significantly increased in the rice bran fed crickets, although with 3.2% of all lipids still trailing far behind the others. What would the differences mean in connection with the health of the consumer (animal or human)?
Response: As your comment, the lipid classes differences in both groups in this study were very slight, however glycolipids in brown rice have functional properties, such as steryl glycosides and glucosylceramide (https://doi.org/10.1016/0005-2760(79)90088-2 and https://doi.org/10.1016/0009-3084(76)90073-6), were reported health benefits from daily consumption. Therefore, we believe that crickets fed rice bran can contribute to health as a source of glycolipids with these functional properties. Also, when manufacturing edible oil, rather than using the total lipids extracted by pressing as they are, further refining processes such as degumming are carried out to remove phospholipids and glycolipids, leaving mostly triglycerides. The phospholipids and glycolipids removed during this process are used as by-products for other purposes.
Comment 9: Although, I’ve listed a number of problems, I would like to see the paper published, provided the authors can accommodate my concerns, cite the missing key papers and once again check their English (already on Line 2 of the Summary it should be “…lipids which contain a…”
Response: Thank you for the comment. We have corrected several grammatical errors in the manuscript.
Comment 10: Incidentally, it seems a little daft to have a “Simple Summary” and an “Abstract”, which basically contain the same information. Ask the journal’s chief editor what s/he wants to have in the Simple Summary and what in the Abstract. To me it’s not clear.
Response: Thank you for the comment. We have corrected the grammatical errors in the manuscript.
Comment 11: you have the occasional problem with the use of "a / an" and "the" as well as singularb and plural. Please check your ms.
Response: Thank you for the comment. We have corrected several grammatical errors in the manuscript.
Reviewer 3 Report
Comments and Suggestions for Authors
Thank you for the opportunity to review your manuscript. While the analysis of cricket oil lipid composition is a relevant topic, the current version requires substantial improvements before it is suitable for publication in this journal. Please carefully address all the points raised in the detailed comments, focusing on enhancing the discussion of your findings in the context of existing literature, providing clear justifications for your methodological choices, elaborating on the implications of your results, and ensuring the overall clarity and accuracy of the manuscript.
Please thoroughly review the entire document for grammatical and orthographic errors. Due to the absence of line numbering, specific locations for these corrections are difficult to pinpoint.
The Discussion section requires significant improvement, particularly in addressing the following points:
The results are not adequately discussed in the identification of lipid classes in cricket oil and their feed compared to existing research. Similarly, Figure 3 lacks sufficient discussion. Please also address the following questions:
- How effective is this method as a preliminary screening tool before more in-depth fat analysis?
- What factors account for the differences observed between the selected solvents?
- Do these initial solvent-based separation results correlate with the subsequent oil analysis findings?
What is the rationale for separating neutral lipids, phospholipids, and glycolipids in the context of distinguishing cricket fats? Beyond their fatty acid profiles, what are their respective roles in the nutritional and health-promoting properties of crickets? Furthermore, if the intention is to incorporate these lipid classes into food formulations, what are their effects on the physicochemical and techno-functional characteristics? What other potential applications could be explored based on their specific lipid class composition? Is this lipid class composition typical for this cricket species, regardless of diet? What advantages does this lipid profile offer compared to other edible insect species? If the intention is to promote these oils as edible sources, please provide more detailed information.
Explain how the ω-6/ω-3 ratio of the rice-bran and control diets influenced the fatty acid composition of both cricket oils. Additionally, discuss the implications of this impact on their potential applications beyond food formulations.
As presented in Table 3, it is necessary to discuss why the fractionation of lipids into classes is advantageous for obtaining fractions rich in SFA, MUFA, or PUFA for various applications. This is particularly relevant given that the differences in these three major fatty acid groups are statistically similar across the different diets.
Comments on the Quality of English LanguageNo comments.
Author Response
Thank you for the opportunity to review your manuscript. While the analysis of cricket oil lipid composition is a relevant topic, the current version requires substantial improvements before it is suitable for publication in this journal. Please carefully address all the points raised in the detailed comments, focusing on enhancing the discussion of your findings in the context of existing literature, providing clear justifications for your methodological choices, elaborating on the implications of your results, and ensuring the overall clarity and accuracy of the manuscript.
Dear Reviewer 3:
Thank you for reviewing our manuscript. All the comments from reviewer 3 are very important and essential for this manuscript. Thank you again for spending your valuable time on this manuscript. We made changes to this manuscript according to suggestions from reviewer 3, and we would you like to confirm the below revised comments and manuscript.
Comment 1: Please thoroughly review the entire document for grammatical and orthographic errors. Due to the absence of line numbering, specific locations for these corrections are difficult to pinpoint.
Response: Thank you for the comment. We have corrected several grammatical errors in the manuscript.
Comment 2: The Discussion section requires significant improvement, particularly in addressing the following points: How effective is this method as a preliminary screening tool before more in-depth fat analysis?
Response: Thank you for the comment. We added the following sentences into the Discussion section(P13, L8 to 12, in revised manuscript).
“In this study, we conducted an analysis of lipid classes and fatty acid composition as a first step toward understanding the effects of rice bran feed on the nutritional components of crickets, because these analyses had been performed in several previous study focused on the effects of cricket feed [19-22].”
Comment 3: What factors account for the differences observed between the selected solvents?
Response: Thank you for the comment. We selected the chloroform, acetone and methanol to separate the neutral lipid, glycolipid and phospholipid. They separate the lipid classes by affinity of the solvent and the substance. And this method was used in other previous studies (doi:10.1016/j.foodchem.2007.03.008 and doi:10.1007/s11746-011-1903-z.) Therefore, we believe this method could sufficiently separate different classes of the lipid. However, we have no doubt about need for more detailed investigation on lipid classes, so we will try to explore more deeply in our future work.
Comment 4: Do these initial solvent-based separation results correlate with the subsequent oil analysis findings?
Response: Thank you for the comment. The results of solvent-based separation are consistent with the results of the TLC analysis, but this type of analysis can’t perform sufficient quantitative analysis, so it is limited to qualitative analysis.
Comment 5: What is the rationale for separating neutral lipids, phospholipids, and glycolipids in the context of distinguishing cricket fats?
Response: Thank you for the comment. We separate these lipid classes to investigate the influence of feed on each lipid class, because many previous studies only focused on the total lipid although significant differences in fatty acid composition depending on lipid class. Also, we hypothesized the influence of feed on cricket lipid may extend beyond fatty acid regeneration to lipid classes.
Comment 6: Beyond their fatty acid profiles, what are their respective roles in the nutritional and health-promoting properties of crickets?
Response: Thank you for the comment. The Abundant fatty acids in the crickets are oleic and linoleic acid. Linoleic acid is one of the essential fatty acids, and oleic acid is known for its ability to inhibit arteriosclerosis.
Comment 7: Furthermore, if the intention is to incorporate these lipid classes into food formulations, what are their effects on the physicochemical and techno-functional characteristics?
Response: Thank you for the comment. The results obtained this time do not allow for in-depth consideration of the points you raised. However, we understand the importance of this point and would like to conduct research on it in the future.
Comment 8: What other potential applications could be explored based on their specific lipid class composition? Is this lipid class composition typical for this cricket species, regardless of diet?
Response: Thank you for the comment. Glycolipids in brown rice have functional properties, such as steryl glycosides and glucosylceramide (https://doi.org/10.1016/0005-2760(79)90088-2 and https://doi.org/10.1016/0009-3084(76)90073-6), were reported health benefits from daily consumption. Therefore, we believe that crickets fed rice bran can contribute to health as a source of glycolipids with these functional properties. And the fatty acid composition is typical for this cricket species.
Comment 9: What advantages does this lipid profile offer compared to other edible insect species?
Response: Thank you for the comment. In general, crickets contain oleic and linoleic acid abundant, and these lipids are also abundant in many other edible insects. Thus, in terms of fatty acid composition, compared to other edible insects, it is considered to have few advantages.
Comment 10: If the intention is to promote these oils as edible sources, please provide more detailed information.
Response:Thank you for the comment. The cricket oils the we used in this study are not sold now.
Comment 11: Explain how the ω-6/ω-3 ratio of the rice-bran and control diets influenced the fatty acid composition of both cricket oils. Additionally, discuss the implications of this impact on their potential applications beyond food formulations.
Response: Thank you for the comment. In this study, the decrease of ω-3 fatty acids in the crickets fed rice bran, and this is thought to be because brown rice does not contain these fatty acids. Unfortunately, there were not any obvious different in the fatty acid composition between both groups, however if the glycolipids that increased in crickets in this study were steryl glycosides, they may have health benefits.
Comment 12: As presented in Table 3, it is necessary to discuss why the fractionation of lipids into classes is advantageous for obtaining fractions rich in SFA, MUFA, or PUFA for various applications. This is particularly relevant given that the differences in these three major fatty acids groups are statistically similar across the different diets.
Response: Thank you for the comment. We think the fractionation method used in this study involves the use of highly toxic solvents such as chloroform and methanol, making it difficult to use industrially for food applications. Therefore, we have decided not to include such points in discussion section.
Reviewer 4 Report
Comments and Suggestions for Authors
- The claimed novelty is weak. Feeding edible insects with agricultural by-products is well-established; using rice bran specifically does not constitute a significant conceptual advance.
- The study presents a variant of known experimental paradigms rather than a novel hypothesis or mechanism.
- The hypothesis is inadequately defined in the introduction. The manuscript implies a relationship between rice bran feed and altered lipid profiles in crickets but fails to explicitly pose a testable hypothesis.
- There is no mechanistic underpinning provided for expecting specific shifts in lipid class or fatty acid distribution. Please fix this.
- The experimental design is superficial. A seven-day feeding duration is insufficient to expect metabolically meaningful changes in insect lipid composition.
- No rationale is provided for the short intervention, and no longitudinal data or time-course validation is presented. The design lacks robustness.
- No advanced profiling (e.g., LC-MS/MS) or lipid oxidation metrics are included, which limits analytical depth.
- Statistical analysis is weak. The sole use of Student's t-tests across numerous comparisons without correction for multiple testing undermines the statistical integrity.
- Furthermore, no multivariate analysis is performed, and effect sizes are missing. Results are statistically marginal and biologically subtle, yet are overinterpreted.
- Sample replication is minimal. Triplicate analyses are reported, but there is no clarification on how many crickets constituted a biological replicate, nor is there information on inter-batch variability. Please fix this.
- Interpretation is overstated. Minor shifts in glycolipid and ω-6/ω-3 ratios are treated as nutritionally significant, despite being small in magnitude and lacking validation through bioactivity assays or health-relevant endpoints.
- No sensory, stability, or functional evaluation is included.
- The study focuses solely on compositional analysis. Without data on taste, shelf-life, oxidative resistance, or physiological impact, claims about functional food potential are unsubstantiated and premature.
- Industrial relevance is limited. There is no discussion of cost, yield, scalability, or compliance with feed regulations, which are critical for evaluating industrial feasibility.
The authors do not address the short feeding period, absence of sensory data, or modest nature of the observed compositional changes. - The concluding remarks imply broad implications that are unsupported by the data. Rewrite the conclusion.
Author Response
Dear Reviewer 4:
Thank you for reviewing our manuscript. All the comments from reviewer 4 are very important and essential for this manuscript. Thank you again for spending your valuable time on this manuscript. We made changes to this manuscript according to suggestions from reviewer 4, and we would you like to confirm the below revised comments and manuscript.
Comment 1: The claimed novelty is weak. Feeding edible insects with agricultural by-products is well-established; using rice bran specifically does not constitute a significant conceptual advance. The study presents a variant of known experimental paradigms rather than a novel hypothesis or mechanism.
Response: Thank you for the comment. We think our study has sufficient novelty, because there are very few studies that have fed crickets with rice bran, and none that have focused on lipid classes and fatty acid composition of crickets in such detail as this study.
Comment 2: The hypothesis is inadequately defined in the introduction. The manuscript implies a relationship between rice bran feed and altered lipid profiles in crickets but fails to explicitly pose a testable hypothesis.
Response: Thank you for the comment. We conducted this study to investigate the influences of feeding crickets with rice bran. And there are little previous studies focused on this agricultural by-product.
Comment 3: There is no mechanistic underpinning provided for expecting specific shifts in lipid class or fatty acid distribution. Please fix this.
Response: Thank you for the comment. The results obtained this time do not allow for in-depth consideration of the points you raised. However, we understand the importance of this point and would like to conduct research on it in the future. And we added the following sentence into discussion section (P14, L14 to 15, in revised manuscript).
“Furthermore, research focusing on such feed will require consideration of costs and yields in order to achieve sustainability.”
Comment 4: The experimental design is superficial. A seven-day feeding duration is insufficient to expect metabolically meaningful changes in insect lipid composition.
Response: Thank you for the comment. We added the following description of the reason we had chosen 7 days for feeding period in Materials and Methods section 2.1. (P3, L28 to 30, in revised manuscript).
“Because in the previous study[17], potential effects to prevent growth and to decrease survival ratio was observed on long-term more than one month feeding of rice bran, we consequently fed cricket with rice bran for only seven days.”
Comment 5: No rationale is provided for the short intervention, and no longitudinal data or time-course validation is presented. The design lacks robustness.
Response: Thank you for the comment. This study was planned based on the design of this previous study (doi:10.3390/foods13111668). In this previous study, significant changes in fatty acid composition and aromatic components were observed after seven days of feeding. However, we think long-term feeding will show different result from this study on the lipid analysis such as lipid class and fatty acid composition, and this is important to understand the underlaying mechanisms in the effect of feed on changes in the nutrition of crickets.
Comment 6: No advanced profiling (e.g., LC-MS/MS) or lipid oxidation metrics are included, which limits analytical depth.
Response: Thank you for the comment. We agree with your opinion. we analyzed only lipid class and fatty acid in this study however, we think the results in this study are very important to understand the influence of by-product feed on lipophilic components of crickets, because there are little knowledges about rearing crickets with broken rice which is regarded as unused resources that are constantly generated in countries that consume rice. We plan to conduct more comprehensive comparative analysis in the future work, including volatile components, lipids and proteins, etc.
Comment 7: Statistical analysis is weak. The sole use of Student's t-tests across numerous comparisons without correction for multiple testing undermines the statistical integrity. Furthermore, no multivariate analysis is performed, and effect sizes are missing. Results are statistically marginal and biologically subtle, yet are overinterpreted.
Response: Thank you for the comment. We changed the statistical analysis method to Welch’s t-test from Student's t-test, according to some previous studies using similar statistical method (doi:10.3390/foods13111668 and https://doi.org/10.3136/fstr.25.597).
Comment 8: Sample replication is minimal. Triplicate analyses are reported, but there is no clarification on how many crickets constituted a biological replicate, nor is there information on inter-batch variability. Please fix this.
Response: Thank you for the comment. We modified as following sentences in Materials and Methods 2.1 Materials (P2, L22 to 24, in revised manuscript), however some information cannot be disclosed as confidential.
“In this study, oils extracted from adult house crickets (Acheta domesticus) by pressing were kindly provided by FUTURENAUT Inc. (Gunma, Japan), a commercial grower. House crickets were reared without separating males and females under two distinct feeding regimes: control feeds and rice brans derived from broken rice. The control feed, commonly used for rearing house crickets, primarily consisted of poultry feed, fish meal powder, skim milk, etc. (Fig. 2 A) [17,26].”
Comment 9: Interpretation is overstated. Minor shifts in glycolipid and ω-6/ω-3 ratios are treated as nutritionally significant, despite being small in magnitude and lacking validation through bioactivity assays or health-relevant endpoints.
Response: Thank you for the comment. We add the following statements about increase of glycolipids into Discussion section (P13, L22 to 23, in revised manuscript).
“However, we couldn’t fully identify the increase glycolipid in this study. Therefore, we believe that more detailed analysis of glycolipids is desirable in the future.”
Comment 10: No sensory, stability, or functional evaluation is included. The study focuses solely on compositional analysis. Without data on taste, shelf-life, oxidative resistance, or physiological impact, claims about functional food potential are unsubstantiated and premature.
Response: Thank you for the comment. We are planning to more comprehensive comparative analysis and in vitro experiment using extracts from both crickets group in the future work. And we would also like to consider positively the evaluation of physical characteristics that you pointed out. We didn't do any sensory evaluation this time because it was hard to gather people, but we did check that the color and flavor were obviously different between the two groups.
Comment 11: Industrial relevance is limited. There is no discussion of cost, yield, scalability, or compliance with feed regulations, which are critical for evaluating industrial feasibility. The authors do not address the short feeding period, absence of sensory data, or modest nature of the observed compositional changes.
Response: Thank you for the comment. The results obtained this time do not allow for in-depth consideration of the points you raised. However, we understand the importance of this point and would like to conduct research on it in the future. And we added the following sentence into discussion section (P14, L14 to 15, in revised manuscript).
“Furthermore, research focusing on such feed will require consideration of costs and yields in order to achieve sustainability.”
Comment 12: The concluding remarks imply broad implications that are unsupported by the data. Rewrite the conclusion.
Response: Thank you for the comment. We have revised our conclusion based on your comments. (P14, L18 to 24, in revised manuscript)
In response to the growing demand for sustainable and nutritionally functional food sources, this study demonstrated that the inclusion of rice bran—derived from an agricultural by-product broken rice—into cricket feed can lead to several notable changes in the lipid composition of house crickets. In this study, we couldn’t find obvious differences from the fatty acid composition between both cricket oils. However, the glycolipid content of cricket oil was significantly increased following rice bran supplementation, without great changes in fatty acid composition.
Reviewer 5 Report
Comments and Suggestions for Authors
The manuscript: Lipid Composition Analysis of Cricket Oil From Crickets Fed With Broken Rice-Derived Bran, presents results or part of results investigating how feeding domestic crickets (Acheta domesticus) with broken rice bran affects the lipid composition of oil extracted from their bodies. The experiment showed that feeding with bran increases the content of glycolipids and phospholipids, and that there are changes in the profile of fatty acids, especially in the ω-6/ω-3 ratio. The importance of this research is great and contributes to the development of sustainable food and functional oils of animal origin through the use of agricultural by-products.
Although the research was adequately conducted, certain shortcomings should be eliminated in accordance with the following comments:
- The feeding period of the crickets referred to in this study was 7 days. It is not clear how he was chosen, and on what basis. Are changes in lipid composition stable or temporary. What is the relation of length of time of diet to lipid metabolism?
- In accordance with the previous comment, the following is also true, although an increase in oleic acid and the ω-6/ω-3 ratio was recorded, the changes in phospho- and glycolipid fractions were insignificant, which reduces the practical significance of the results, and practically does not provide convincing data to conclude that diet has an effect on the composition of the lipid fractions of cricket oil. A brief explanation of this situation is in order.
- In the research, a comparative analysis of the diet with Broken Rice-Derived Bran was performed in relation to the control type of diet. In this case, there are certain ambiguities that need to be cleared up. It is not completely clearly described what the control type of diet is, whether it is something common, standard, etc. Which is why a diet based on the presence of Broken Rice-Derived Bran was chosen for the study. What were the expectations, and why were some other types of nutrition not examined, e.g. with some possible by-products for feeding the crickets?
- Although the analysis of fatty acid composition by GC-FID was described in detail in previous publications [32,33], this manuscript should still include at least one sentence on the method of identifying fatty acids (which standard, etc.?), as well as on the method of their quantitative determination.
Author Response
The manuscript: Lipid Composition Analysis of Cricket Oil From Crickets Fed With Broken Rice-Derived Bran, presents results or part of results investigating how feeding domestic crickets (Acheta domesticus) with broken rice bran affects the lipid composition of oil extracted from their bodies. The experiment showed that feeding with bran increases the content of glycolipids and phospholipids, and that there are changes in the profile of fatty acids, especially in the ω-6/ω-3 ratio. The importance of this research is great and contributes to the development of sustainable food and functional oils of animal origin through the use of agricultural by-products.
Dear Reviewer 5:
Thank you for reviewing our manuscript. All the comments from reviewer 5 are very important and essential for this manuscript. Thank you again for spending your valuable time on this manuscript. We made changes to this manuscript according to suggestions from reviewer 5, and we would you like to confirm the below revised comments and manuscript.
Comment 1:
The feeding period of the crickets referred to in this study was 7 days. It is not clear how he was chosen, and on what basis.
Response:
Thank you for the comment. We added the following description of the reason we had chosen 7 days for feeding period in Materials and Methods section 2.1. (P3, L28 to 30, in revised manuscript).
“Because in the previous study[17], potential effects to prevent growth and to decrease survival ratio was observed on long-term more than one month feeding of rice bran, we consequently fed cricket with rice bran for only seven days.”
Comment 2:
Are changes in lipid composition stable or temporary. What is the relation of length of time of diet to lipid metabolism?
Response:
Thank you for the insightful comment. We think these slightly changes on lipid classes composition and fatty acid composition are temporary, because crickets can not synthesize ω-3 fatty acids and cholesterols by de novo pathway. Therefore, we think that several trends observed in our study such as increase in oleic acid and the ω-6/ω-3 ratio are due to rice bran diet. However, it is still unknown how long these changes on their lipid classes and fatty acid composition, and how much impact rice bran has on the cricket lipid metabolism. Therefore, we think that we should investigate these points in the future work.
Comment 3:
In accordance with the previous comment, the following is also true, although an increase in oleic acid and the ω-6/ω-3 ratio was recorded, the changes in phospho- and glycolipid fractions were insignificant, which reduces the practical significance of the results, and practically does not provide convincing data to conclude that diet has an effect on the composition of the lipid fractions of cricket oil. A brief explanation of this situation is in order.
Response:
As your comment, Oleic acid didn’t significantly change in phospho- and glycolipid fractions, but the ω-6/ω-3 ratio significantly changed in these fractions. Moreover, oleic acid is most abundant fatty acid in both diets, in addition most absorbed lipid are generally stored in the form of triglycerides. For these reasons, it is possible that there was little influence on phospholipids and glycolipids.
Comment 4:
In the research, a comparative analysis of the diet with Broken Rice-Derived Bran was performed in relation to the control type of diet. In this case, there are certain ambiguities that need to be cleared up. It is not completely clearly described what the control type of diet is, whether it is something common, standard, etc.
Response:
Thank you for your valuable comment. Control diet in this study consists of commercially available poultry feed primarily, also contains fish meal powder, skim milk, etc. This type of diet is used not only for commercial rearing, but for experiments in many previous studies which we have already referred on our manuscript. And we added the following sentence into Materials and Methods section (P3, L21 to 22, in revised manuscript).
“The control feed, commonly used for rearing house crickets, primarily consisted of poultry feed, fish meal powder, skim milk, etc. (Fig. 2 A) [17,26].”
Comment 5:
Which is why a diet based on the presence of Broken Rice-Derived Bran was chosen for the study. What were the expectations, and why were some other types of nutrition not examined, e.g. with some possible by-products for feeding the crickets?
Response:
The purpose of our study is to determine the influence of rice-bran on the composition of lipid class and fatty acid. As mentioned above, we had expected that some significant differences on fatty acid composition even with short-term feeding of 7 days, because in the previous study that feed the crickets with apple byproduct for same period with our study, several significant differences were observed on fatty acid composition. However, in our study, those differences are slight even it is significant, these relationships between feed ingredients and feeding period should be examined in our future study.
Comment 6:
Although the analysis of fatty acid composition by GC-FID was described in detail in previous publications [32,33], this manuscript should still include at least one sentence on the method of identifying fatty acids (which standard, etc.?), as well as on the method of their quantitative determination.
Response:
Thank you for the valuable comment. We added following sentence into Materials and Methods section 2.5. Analysis of fatty acid composition by GC-FID (P5, L21 to 23, in revised manuscript).
“and heptadecanoic acid was used as an internal standard. Each fatty acids were identified and calculated ratio by FAME-Mix (Sigma Aldrich, St. Louis, USA).”

Round 2
Reviewer 2 Report
Comments and Suggestions for Authors
Nice manuscript, much improved. However, the references [42, 43, 44] cannot be fopund! Please insert then in the appropriate places AND please correct [42], an important citation, of which you are citing the authors' names wrongly. They should be:
Meyer-Rochow VB, Gahukar RT, Ghosh S, Jung C.
Also, with regard to stearic acid as one of the saurated fatty acids shown to have a positive effect on human health, I'd like to see this paper cited:
Bonanome, A., Grundy, S.M., 1988. Effect of dietary stearic acid on plasma cholesterol and lipoprotein levels. N. Engl. J. Med. 318, 1244–1248.
Author Response
Comments from reviewer2:
Comments and Suggestions for Authors
Nice manuscript, much improved. However, the references [42, 43, 44] cannot be fopund! Please insert then in the appropriate places AND please correct [42], an important citation, of which you are citing the authors' names wrongly. They should be:
Meyer-Rochow VB, Gahukar RT, Ghosh S, Jung C.
Also, with regard to stearic acid as one of the saurated fatty acids shown to have a positive effect on human health, I'd like to see this paper cited:
Bonanome, A., Grundy, S.M., 1988. Effect of dietary stearic acid on plasma cholesterol and lipoprotein levels. N. Engl. J. Med. 318, 1244–1248.
Reply:
Thank you very much for your positive feedback and for pointing out the issues with the references.Following your suggestions, we have carefully revised the references and made the following changes.
We have inserted the missing references [43, 44] in the appropriate places within the manuscript to ensure consistency between in-text citations and the reference list.
We have corrected reference [42], now cited as [13], and updated the authors’ names to the correct format
Meyer-Rochow, V.B.; Gahukar, R.T.; Ghosh, S.; Jung, C. Chemical Composition, Nutrient Quality and Acceptability of Edible Insects Are Affected by Species, Developmental Stage, Gender, Diet, and Processing Method. Foods 2021, 10, 1036. doi: 10.3390/foods10051036
Additionally, in accordance with your recommendation regarding stearic acid, we have added the following citation as [45].
Bonanome A.; Grundy S.M. Effect of Dietary Stearic Acid on Plasma Cholesterol and Lipoprotein levels. N Engl J Med 1988, 318, 1244–1248. doi: 10.1056/NEJM198805123181905
We believe that these revisions improved the accuracy and completeness of the references and enhance the overall quality of the manuscript. We very very sincerely appreciate your careful review and valuable comment.
Reviewer 3 Report
Comments and Suggestions for Authors
I acknowledge the considerable effort made in addressing my previous comments. The manuscript has been substantially improved, and the revisions have strengthened both the scientific clarity and overall quality of the work. My only remaining observation is that the order of the references and in-text citations still appears inconsistent and should be carefully revised to ensure they follow the journal’s guidelines. Once this issue is corrected, I consider the manuscript ready for publication.
Author Response
Comment from Reviewer 3:
Comments and Suggestions for Authors
I acknowledge the considerable effort made in addressing my previous comments. The manuscript has been substantially improved, and the revisions have strengthened both the scientific clarity and overall quality of the work. My only remaining observation is that the order of the references and in-text citations still appears inconsistent and should be carefully revised to ensure they follow the journal’s guidelines. Once this issue is corrected, I consider the manuscript ready for publication.
Reply:
Thank you very much for your careful review and your positive feedback on our revised manuscript.
Regarding your remaining observation on the inconsistency in the order of the references and in-text citations, we carefully revised them to ensure full compliance with the journal’s guidelines.
We are deeply grateful for your valuable time, effort, and constructive suggestions throughout the review process.
Reviewer 4 Report
Comments and Suggestions for Authors
The authors have substantially revised manuscript; therefore, it can be accepted.
Author Response
Comments from rviewer 4:
Comments and Suggestions for Authors
The authors have substantially revised manuscript; therefore, it can be accepted.
Reply:
Thank you very much for your time and effort in reviewing our revised manuscript.
We are grateful for your positive evaluation and for recommending our manuscript for acceptance.
We appreciate your support and constructive feedback throughout the review process.